# Context-dependent reversal of odorant preference is driven by inversion of the response in a single sensory neuron type

**Munzareen Khan**[1], **Anna H. Hartmann**[1¤a], **Michael P. O'Donnell**[1¤b], **Madeline Piccione**[1], **Anjali Pandey**[1], **Pin-Hao Chao**[1¤c], **Noelle D. Dwyer**[2], **Cornelia I. Bargmann**[3], **Piali Sengupta** [1]*

1 Department of Biology, Brandeis University, Waltham, Massachusetts, United States of America,
2 Department of Cell Biology, University of Virginia, Charlottesville, Virginia, United States of America,
3 The Rockefeller University, New York, New York, United States of America

¤a Current address: Department of Neurobiology, Harvard Medical School, Boston, Massachusetts, United States of America
¤b Current address: Department of Molecular, Cellular and Developmental Biology, Yale University, New Haven, Connecticut, United States of America
¤c Current address: Deallus Consulting, New York, New York, United States of America
* sengupta@brandeis.edu

**Data Availability Statement:** Numerical data for all behavioral and calcium imaging experiments shown in Figs 1–4 and S1–S5 are provided in https://doi.org/10.5281/zenodo.6537728. All other

## Abstract

The valence and salience of individual odorants are modulated by an animal's innate preferences, learned associations, and internal state, as well as by the context of odorant presentation. The mechanisms underlying context-dependent flexibility in odor valence are not fully understood. Here, we show that the behavioral response of *Caenorhabditis elegans* to bacterially produced medium-chain alcohols switches from attraction to avoidance when presented in the background of a subset of additional attractive chemicals. This context-dependent reversal of odorant preference is driven by cell-autonomous inversion of the response to these alcohols in the single AWC olfactory neuron pair. We find that while medium-chain alcohols inhibit the AWC olfactory neurons to drive attraction, these alcohols instead activate AWC to promote avoidance when presented in the background of a second AWC-sensed odorant. We show that these opposing responses are driven via engagement of distinct odorant-directed signal transduction pathways within AWC. Our results indicate that context-dependent recruitment of alternative intracellular signaling pathways within a single sensory neuron type conveys opposite hedonic valences, thereby providing a robust mechanism for odorant encoding and discrimination at the periphery.

## Introduction

Organisms live in complex and dynamic chemical environments. Animals continuously encounter heterogenous mixtures of chemicals that fluctuate in their concentrations and temporal properties, and provide information about the presence and location of food, mates, competitors, and predators [1,2]. A particularly critical task of the olfactory system is to

data are within the paper and its Supporting Information files.

**Funding:** This work was supported in part by the National Science Foundation (www.nsf.gov) (IOS 165518 and IOS 2042100 – P.S., MRSEC 2011846 to Brandeis University), and the National Institutes of Health (www.nih.gov) (R35 GM122463 - P.S., F31 DC015186 - A.H.H.). The funders had no role in study design, data collection and analysis, decision to publish, or preparation of the manuscript.

**Competing interests:** The authors have declared that no competing interests exist.

**Abbreviations:** cGMP, cyclic guanosine monophosphate; IAA, isoamyl alcohol; sIAA, saturating IAA.

discriminate among related chemical cues. Since chemicals that are structurally related can have distinct saliences for an organism [3], these cues must be identified and differentiated in order to elicit the appropriate behavior. In particular, animals need to distinguish individual chemicals in a complex mixture, or detect the presence of a new chemical in the background of a continuously present odorant (e.g., [4–7]). Context-dependent odorant discrimination can be driven via integration and processing of multiple sensory inputs in central brain regions (e.g., [8–13]). Although mechanisms operating at the level of single sensory neuron types or sensilla in the periphery have also been implicated in this process [14–21], the contributions of sensory neurons to mediating odorant discrimination and olfactory behavioral plasticity are not fully understood.

*Caenorhabditis elegans* senses and navigates its complex chemical environment using a small subset of sensory neurons [22–24]. The valence of individual chemicals is largely determined by the responding sensory neuron type, such that distinct subsets of chemosensory neurons drive either attraction or avoidance to different chemicals [25,26]. Each chemosensory neuron type in *C. elegans* expresses multiple chemoreceptors that are likely tuned to different odorants, a subset of which can be behaviorally discriminated [27,28]. Failure to discriminate between 2 attractive odorants sensed by a single chemosensory neuron results in loss or reduced attraction, but not aversion, to the test chemical [25]. In contrast, aversion of a normally attractive chemical occurs as a consequence of modulation by experience and internal state [29–35]. Experience- and state-dependent switches in the valence of a chemical are mediated largely via integration and plasticity at the synaptic and circuit level [36–41], and have not been reported to be driven solely by plasticity in sensory neuron responses.

Here, we report that a context-dependent inversion of the response to medium-chain alcohols in the AWC sensory neuron pair in *C. elegans* switches the valence of the behavior from attraction to aversion. We show that adult *C. elegans* hermaphrodites are attracted to low concentrations of bacterially produced medium-chain alcohols such as 1-hexanol and 1-heptanol (henceforth referred to as hexanol and heptanol, respectively). However, when presented in the context of a uniform saturating background of a subset of other attractive odorants also sensed by AWC, animals are now strongly repelled by these alcohols. Hexanol and heptanol inhibit AWC in nonsaturating conditions to drive attraction, but in saturating odor conditions, these chemicals instead activate AWC to promote avoidance. We show that this context-dependent response inversion is mediated by distinct downstream intracellular effectors including Gα proteins and receptor guanylyl cyclases in AWC. Results described here indicate that context-dependent engagement of distinct intracellular signaling pathways within a single sensory neuron type is not only sufficient to discriminate between structurally related chemicals, but also to convey opposing hedonic valences.

## Results

### Medium-chain alcohols can either attract or repel *C. elegans* based on odorant context

Cross-saturation assays have previously been used as a measure of the ability of *C. elegans* to behaviorally discriminate between 2 volatile odorants [25]. In these assays, animals are challenged with a point source of the test chemical in the presence of a uniform concentration of a second saturating chemical (Fig 1A). The ability to retain responses to the test chemical under these conditions indicates that the animal is able to discriminate between the test and saturating odorants.

The AWC olfactory neuron pair in *C. elegans* responds to low concentrations of a subset of bacterially produced attractive chemicals including benzaldehyde, isoamyl alcohol (IAA), and

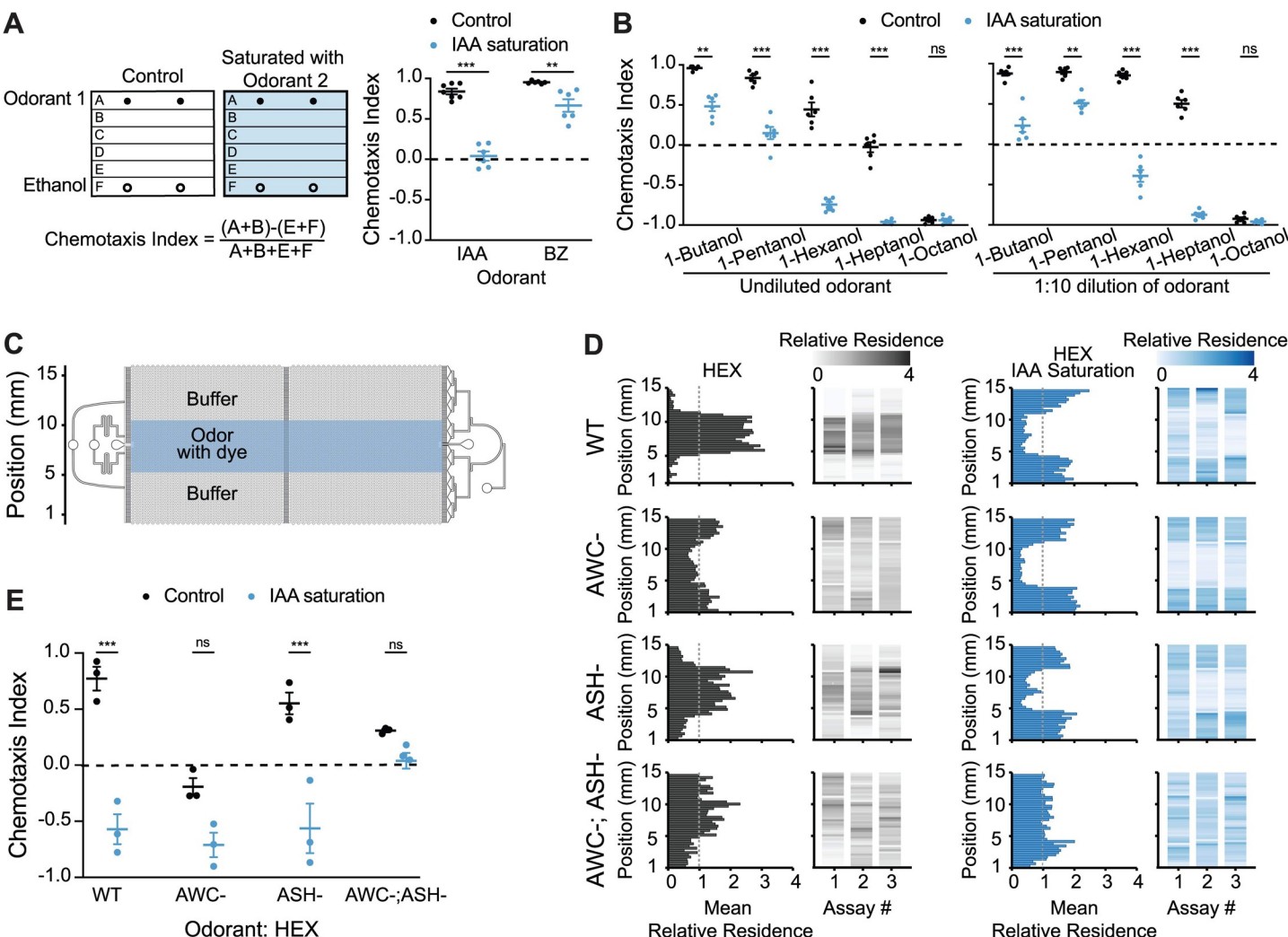

**Fig 1. The AWC neurons drive attraction or avoidance of hexanol in the absence or presence of sIAA, respectively. (A)** (Left) Cartoon of plate chemotaxis assay (see Materials and methods). Filled and open circles: location of 1 μl each of the test odorant or ethanol. Positive and negative chemotaxis indices indicate attraction and avoidance, respectively. (Right) Behaviors of wild-type animals on control or sIAA plates. Test odorants: 1:200 dilution of BZ, 1:1,000 dilution of IAA. **(B)** Behaviors of wild-type animals on control and sIAA plates containing the indicated dilutions of different alcohols as the test odorants. **(C)** Schematic of the microfluidics behavioral device used in behavioral assays. Adapted from [42]. **(D)** Average histograms showing mean relative $x$-$y$ residence (relative to spatial odor pattern) of animals of the indicated genotypes over 20 minutes in devices with a central stripe of $10^{-4}$ HEX without (left), or with, a uniform concentration of $10^{-4}$ IAA (right) in the device. Indicated neurons in each examined strain were genetically ablated via the expression of caspases (S1 Table). Mean relative residence >1 or <1 (dashed vertical line) indicate attraction and avoidance, respectively. Corresponding heat maps show the density of tracks in the $y$-position for each assay. $n = 20$–30 animals per assay; 3 biological replicates. **(E)** Chemotaxis indices calculated from behavioral assays shown in D. In A, B, each dot is the chemotaxis index of a single assay plate containing approximately 100–200 adult hermaphrodites. Assays were performed in duplicate over at least 3 days. In E, each dot is the chemotaxis index from a single assay in behavior chips. Long horizontal bars indicate the mean; errors are SEM. **, ***: $P < 0.01$ and 0.001, respectively (A, B: Kruskal–Wallis with post hoc pairwise Wilcoxon test and Benjamini–Hochberg method for $P$-value correction; E: 2-way ANOVA with Bonferroni's correction); ns, not significant. Underlying data are provided in https://doi.org/10.5281/zenodo.6537728. BZ, benzaldehyde; HEX, hexanol; IAA, isoamyl alcohol; sIAA, saturating IAA.

the short-chain alcohol 1-pentanol [25,41,43]. In the presence of a uniform background concentration of IAA, animals are indifferent to a point source of IAA, but retain the ability to respond to a point source of the structurally distinct chemical benzaldehyde (Fig 1A) [25]. To investigate the extent to which AWC is able to discriminate among structurally related chemicals, we examined responses to point sources of straight-chain alcohols with or without saturating IAA (henceforth referred to as sIAA). Attractive responses to the short-chain alcohols

1-butanol and 1-pentanol were reduced or abolished in sIAA indicating that animals are largely unable to discriminate between these chemicals (Fig 1B) [25]. *C. elegans* is robustly repelled by long-chain alcohols such as 1-octanol [25]; this response is mediated by integration of sensory inputs from multiple sensory neurons including AWC in a food-dependent manner [27,44,45]. sIAA did not affect avoidance of 1-octanol although we note that avoidance of this chemical is maximal even in control conditions (Fig 1B). These observations indicate that in sIAA, AWC-driven attractive responses to a subset of related alcohols primarily sensed by AWC is decreased or abolished, but that long-range avoidance of alcohols driven by additional sensory neurons may be unaffected.

Animals were also attracted to the medium-chain alcohols hexanol and heptanol (Fig 1B) [25]. Unlike the reduced attraction observed for short-chain alcohols in sIAA, animals in sIAA strongly avoided both hexanol and heptanol (Fig 1B). In contrast, saturation with either hexanol or heptanol abolished attraction to a point source of IAA but did not result in avoidance (S1A Fig). Many chemicals elicit distinct behaviors at different concentrations (e.g., [25,46–49]). Similarly, worms were attracted to low, but were weakly repelled by high, concentrations of hexanol (S1B Fig). The response was shifted towards indifference of lower, and strong avoidance of higher, hexanol concentrations in sIAA (S1B Fig). We infer that distinct underlying antagonistic neuronal pathways mediate attraction to, and avoidance of, hexanol at all concentrations. Attraction predominates at lower, and avoidance at higher, hexanol concentrations. sIAA appears to inhibit the attraction pathway thereby promoting a shift towards hexanol avoidance at all concentrations. These results also indicate that animals are unable to discriminate between IAA and hexanol for attraction but are able to do so for avoidance.

## The attraction-promoting AWC olfactory neuron pair instead drives hexanol avoidance in odorant saturation conditions

The ability of hexanol and IAA to cross-saturate for attraction suggested that hexanol attraction is also mediated by AWC. The AWB, ASH, and ADL sensory neuron pairs mediate avoidance of noxious alcohols including high concentrations of IAA [26]. We tested whether hexanol attraction and avoidance require AWC and one or more of the avoidance-mediating sensory neurons, respectively.

To more reliably characterize the contributions of different sensory neurons to hexanol attraction and avoidance behaviors, we used microfluidics behavioral arenas (Fig 1C) that enable precise spatiotemporal control of stimulus delivery together with automated tracking of individual worm movement [42]. Animals were distributed throughout the device in buffer alone, but accumulated within a spatially restricted stripe of IAA or hexanol over the assay period indicating attraction (Figs 1D and 1E and S1C and S1D and S1 Video). In the presence of a uniform low concentration of IAA throughout the device, animals were indifferent to a central stripe of IAA indicating response saturation (S1C and S1D Fig). In contrast, in these sIAA conditions, animals avoided the central hexanol stripe (Fig 1D and 1E and S1 Video). Thus, the behaviors in microfluidics devices recapitulate the behavioral responses observed in plate chemotaxis assays.

Animals in which AWC was either genetically ablated via the expression of caspases, or silenced via the expression of a gain-of-function allele of the *unc-103* potassium channel (S1 Table) [50], were no longer robustly attracted to hexanol but instead exhibited weak avoidance, indicating that AWC is necessary for hexanol attraction (Figs 1D and 1E and S1E). AWC-ablated animals also failed to be attracted to the AWC-sensed odorant benzaldehyde but were not repelled by this chemical [25] (S1F Fig). In sIAA, AWC-ablated or silenced animals

continued to robustly avoid hexanol suggesting that this aversion is mediated by a neuron type other than AWC (Figs 1D and 1E and S1E). Genetic ablation of the nociceptive neuron type ASH resulted in a failure to avoid high concentrations of the ASH-sensed chemical glycerol (S1G Fig) [51], but had only minor effects on attraction to, or avoidance of, hexanol without or with sIAA, respectively (Fig 1D and 1E). However, animals in which both AWC and ASH were genetically ablated were indifferent to hexanol regardless of conditions (Fig 1D and 1E). We conclude that while AWC drives attraction to hexanol, either AWC or ASH can mediate hexanol avoidance in sIAA. Thus, the typically attraction-mediating AWC sensory neuron pair is able to drive hexanol avoidance based on odorant context.

## Saturation with AWC-sensed odorants cell-autonomously inverts the hexanol response in AWC

To investigate the mechanisms by which AWC mediates hexanol attraction or avoidance in a context-dependent manner, we next examined hexanol-evoked changes in intracellular calcium dynamics. Attractive odorants decrease intracellular calcium levels in AWC [43]. Odorant-mediated inhibition of AWC as well as disinhibition and/or activation upon odorant removal together drive attraction [43,52]. We first confirmed that the transgenic strain expressing the genetically encoded calcium indicator GCaMP3 in AWC exhibited behavioral responses to hexanol similar to those of wild-type animals (S2A Fig). We next assessed AWC calcium dynamics in animals immobilized in microfluidics imaging chips [53] in response to hexanol and sIAA concentrations that elicited attraction and avoidance behaviors in microfluidics behavioral arenas (Fig 1D and 1E).

As reported previously [43], low IAA concentrations that robustly attract wild-type animals decreased, and removal increased, intracellular calcium levels in AWC (S2B Fig). In sIAA, an additional pulse of IAA led to only a further minor decrease in calcium levels in AWC upon odor onset (S2B Fig), correlating with loss of attraction to IAA in these saturation conditions (S1C and S1D Fig). Addition of low hexanol concentrations that attracts wild-type animals also decreased calcium levels in AWC (Fig 2A and 2B and S2 Video). However, in sIAA, hexanol instead robustly increased intracellular calcium in AWC correlating with hexanol avoidance under these conditions (Figs 2A and 2B and S2C and S2 Video). The inversion in the sign of the hexanol-evoked response in sIAA was also observed using higher and lower hexanol concentrations (S2D Fig). Neither a hexanol/IAA mixture, nor preexposure to IAA followed by a hexanol pulse, was sufficient to elicit an increase in calcium levels in AWC (S2E and S2F Fig). Moreover, upon saturation with hexanol, IAA elicited variable responses of smaller amplitude, but did not invert the response (S2G Fig), consistent with animals being indifferent to IAA in hexanol saturation conditions (S1A Fig).

Animals also avoid heptanol in sIAA (Fig 1B); the heptanol response in AWC was also inverted in this context (Fig 2C). The AWC neuron pair exhibits bilateral response asymmetry to a subset of odorants [43,54,55]. Approximately 90% of imaged AWC neurons responded to both hexanol and heptanol with and without sIAA (Fig 2A and 2C), implying that both AWC neurons are able to respond to these chemicals regardless of conditions. Hexanol-driven calcium increases and decreases in AWC were maintained in animals carrying *unc-13* and *unc-31* mutant alleles that lack synaptic and peptidergic transmission, respectively [56,57] (Figs 2B and S2H), suggesting that these responses are mediated cell-autonomously. However, we are unable to exclude the possibility that these responses are modulated in part cell nonautonomously via gap junctions.

To assess the extent to which saturation with 1 AWC-sensed odorant influences the response to a pulse of a second AWC-sensed chemical, we next examined responses to

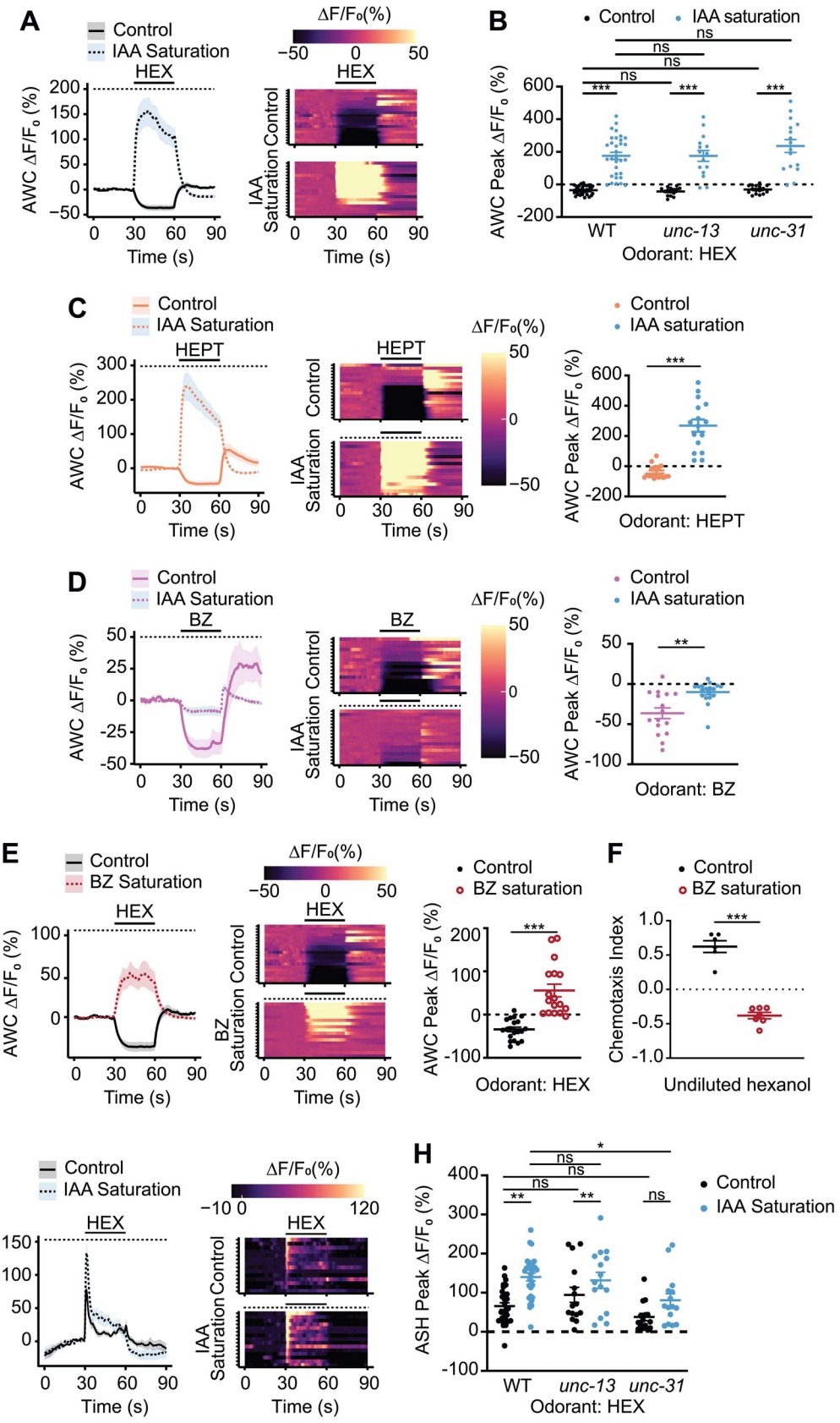

**Fig 2. Hexanol-mediated inhibition of AWC is inverted to activation in saturating odor conditions. (A, C, D, E, G)** (Left) Average changes in GCaMP3 fluorescence in AWC (A, C, D, E) and ASH (G) in response to a 30-second pulse of $10^{-4}$ dilution of the indicated odorant(s) (solid line). The presence of saturating chemicals in the imaging chip at $10^{-4}$ dilution is indicated by a dashed line. Shaded regions are SEM. (Right in A, G, Center in C, D, E) Corresponding heatmaps of changes in fluorescence intensity. Each row in the heatmaps shows responses from a single AWC (A, C, D, E) or ASH (G) neuron from different animals; $n \geq 15$ each. (Right in C, D, E) Quantification of peak fluorescence intensity changes upon odorant onset under nonsaturated or saturated conditions. Each dot is the response from a single neuron. Wild-type hexanol response data in control conditions were interleaved with experimental data in A, E and are repeated. Raw traces of hexanol-evoked responses in AWC in control and sIAA conditions are shown in S2C Fig. **(B, H)** Quantification of peak fluorescence intensity changes upon odorant onset under nonsaturated or saturated conditions in AWC (B) and ASH (H). Each dot is the response from a single neuron. Wild-type data are from A (for B) and G (for H). Average traces and heatmaps of responses in *unc-13(e51)* and *unc-31(e928)* mutants in AWC and ASH are shown in S2H and S4C and S4D Figs, respectively. **(F)** Behavioral responses of animals to a point source of undiluted hexanol on plates with or without saturating benzaldehyde at $10^{-4}$ dilution. Each dot is the chemotaxis index of a single assay plate containing approximately 100–200 adult hermaphrodites. Assays were performed in duplicate over at least 3 days. Long horizontal bars indicate the mean; errors are SEM. *, **, ***: $P < 0.05$, 0.01, and 0.001 and, respectively (C–F: Mann–Whitney–Wilcoxon test; B, H: Kruskal–Wallis with post hoc pairwise Wilcoxon test and Benjamini–Hochberg method for *P*-value correction); ns, not significant. Underlying data for this figure are provided in https://doi.org/10.5281/zenodo.6537728. BZ, benzaldehyde; HEPT, 1-heptanol; HEX, 1-hexanol; IAA, isoamyl alcohol; sIAA, saturating IAA.

different odorant/saturating odorant combinations in AWC. We first tested whether the observed sIAA-dependent inversion of the response to hexanol and heptanol extends to additional AWC-sensed odorants. However, in sIAA, benzaldehyde continued to decrease calcium levels in AWC albeit with a significantly reduced amplitude than in control conditions (Fig 2D). These responses are consistent with decreased attraction to, but not avoidance of, a point source of benzaldehyde in sIAA (Fig 1A). Calcium levels in AWC were also robustly decreased by the attractive chemicals diacetyl and pyrazine (S3A and S3B Fig). In sIAA, we again observed odorant-evoked inhibition with decreased amplitude, but not activation of AWC (S3A and S3B Fig). Attraction to pyrazine was unaltered in sIAA (S3C Fig), likely due to this behavior being driven primarily by the AWA olfactory neuron pair [25]. These results indicate that in sIAA, the observed response inversion upon odorant addition to AWC may be elicited only by specific chemicals such as hexanol and heptanol.

We next tested how saturation by an AWC-sensed odorant other than IAA affects hexanol responses in AWC. Saturation with benzaldehyde resulted in hexanol-driven increases in calcium in AWC together with avoidance of hexanol similar to sIAA (Fig 2E and 2F). However, we did not observe similar hexanol-mediated activation of AWC in either saturating diacetyl or pyrazine (S3D and S3E Fig). We conclude that the observed inversion of the response sign in sIAA in AWC is largely specific to hexanol and likely heptanol. Moreover, hexanol-evoked activation of AWC is restricted to saturation by only a subset of AWC-sensed odorants (summarized in S3F Fig; see Discussion).

## ASH responds similarly to hexanol in the absence or presence of a saturating odor

Since our behavioral experiments indicate that the ASH neurons also contribute to hexanol avoidance (Fig 1D and 1E), we also examined hexanol-evoked responses in ASH. We verified that the transgenic strain expressing GCaMP3 in ASH exhibited behavioral responses to hexanol similar to those of wild-type animals (S4A Fig). Hexanol elicited a robust phasic response in ASH neurons with a rapid and transient rise upon hexanol addition (Fig 2G) [58,59]. Response dynamics were similar in sIAA, although the response peak as well as the response baseline in the presence of hexanol were consistently higher as compared to the responses in control conditions (Figs 2G and 2H and S4B).

Hexanol responses in ASH in *unc-13* mutants specifically defective in synaptic transmission [56] resembled those in wild-type animals (Figs 2H and S4C). However, in *unc-31* mutants defective in neuropeptidergic signaling [56,57], only a subset of animals responded to hexanol in control conditions although a larger fraction responded in sIAA (Figs 2H and S4D). Moreover, the dynamics of the hexanol response in sIAA were altered such that the response appeared to be tonic (S4D Fig). We tested whether peptidergic signaling from AWC might modulate hexanol response dynamics in ASH. Due to technical limitations, we could not assess hexanol responses in ASH in AWC-ablated animals. Sensory responses in AWC (including hexanol responses, see Fig 4A) are abolished in animals mutant for the *tax-4* cyclic nucleotide-gated channel [25,60]; this channel subunit gene is not expressed in ASH [60]. Hexanol responses in ASH in *tax-4* mutants resembled those in *unc-31* mutants (S4E Fig), suggesting that AWC or another *tax-4*-expressing neuron likely influences hexanol response dynamics in ASH. Together, these results indicate that unlike our observations in AWC, the hexanol response in ASH is not inverted in sIAA. However, while ASH likely responds cell-autonomously to hexanol, these responses may be modulated by AWC or possibly another *tax-4*-expressing neuron.

## The ODR-3 Gα protein is necessary for the hexanol-driven response inversion in AWC in odorant saturation conditions

To probe the molecular mechanisms underlying the context-mediated hexanol response inversion in AWC, we next tested the behaviors and hexanol responses of mutants previously implicated in AWC sensory signal transduction. Binding of odorants to their cognate receptors in AWC decreases intracellular cyclic guanosine monophosphate (cGMP) levels either via inhibiting the activity of receptor guanylyl cyclases such as ODR-1 and DAF-11, and/or by promoting the activity of one or more phosphodiesterases, via heterotrimeric G proteins (Fig 3A) [26]. Reduced intracellular cGMP levels closes cGMP-gated channels, inhibits calcium influx, and promotes attraction (Fig 3A) [26,43]. The inversion in the hexanol response in AWC in control and sIAA conditions indicates that hexanol likely acts via distinct molecular mechanisms in AWC under different conditions to evoke a response.

While the hexanol and IAA receptors in AWC are unknown, multiple Gα proteins are expressed in AWC and have been implicated in mediating odorant signal transduction in this neuron type (Fig 3A) [26,62–64]. Loss of function mutations in the ODR-3 $G\alpha_i$/$G\alpha_o$-like protein decreases although does not fully abolish attraction to multiple AWC-sensed chemicals including IAA [62,64–66]. *odr-3* null mutants continued to be attracted to hexanol, indicating that this Gα protein is partly dispensable for this behavior (Figs 3B and 3C and S5A). Animals mutant for the additional AWC-expressed nematode-specific Gα genes *gpa-2*, *gpa-3*, and *gpa-13*, as well as animals triply mutant for all 3 *gpa* genes, also retained the ability to be attracted to hexanol (S5A Fig). In contrast, *odr-3*, but not the *gpa* single or triple mutants, no longer avoided hexanol in sIAA, but were instead attracted, similar to the behaviors of these animals under control conditions (Figs 3B and 3C and S5A). *gpa-3 gpa-13 odr-3* triple mutants were also attracted to hexanol in sIAA (S5A Fig). Hexanol avoidance behavior was rescued upon expression of *odr-3* specifically in AWC (Fig 3B and 3C).

To correlate neuronal responses with behavior, we next examined hexanol-evoked intracellular calcium dynamics in AWC in *odr-3* mutants. While hexanol decreased intracellular calcium concentrations in AWC similarly in both wild-type and *odr-3* animals, hexanol failed to increase calcium levels in AWC in sIAA in *odr-3* mutants (Fig 3D and 3E). Instead, hexanol continued to inhibit AWC in these animals (Figs 3D and 3E and S5C) consistent with *odr-3* mutants retaining attraction to hexanol in sIAA. A possible explanation for the observed

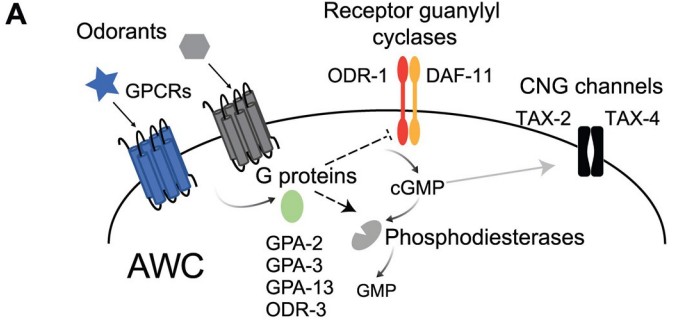

**Fig 3. The ODR-3 Gα protein is required for hexanol-mediated activation of AWC in sIAA. (A)** Cartoon of the olfactory signal transduction pathway in AWC. Adapted from [61]. **(B)** Average residence histograms and heatmaps as described for Fig 1D. *n* = 20–30 animals per assay; 3 biological replicates. An *odr-3* cDNA was expressed in AWC under the *odr-1* promoter. The concentration of hexanol was $10^{-4}$. **(C)** Chemotaxis indices calculated from behavioral assays shown in B. Each dot is the chemotaxis index from a single assay in behavior chips. **(D)** (Left) Average changes in GCaMP3 fluorescence in AWC in response to a pulse of $10^{-4}$ hexanol in wild-type and *odr-3(n2150)* mutants. The presence of saturating chemicals in the imaging chip at $10^{-4}$

dilution is indicated by a dashed line. Shaded regions are SEM. (Right) Corresponding heatmaps of changes in fluorescence intensity; $n \geq 15$ each. A subset of wild-type hexanol response data in control conditions were interleaved with data in Figs 2A and 2E and 4C and are repeated; wild-type hexanol response data in sIAA were interleaved with experimental data in Fig 4C and are repeated. Raw traces of hexanol-evoked responses in AWC in control and sIAA conditions are shown in S5C Fig. (E) Quantification of fluorescence intensity changes upon hexanol odorant onset under nonsaturated or saturated conditions from data shown in D. Each dot is the response from a single neuron. Long horizontal bars indicate the mean; errors are SEM. *, ***: $P < 0.05$ and 0.001, respectively (C: 2-way ANOVA with Bonferroni's correction; E: Kruskal–Wallis with post hoc pairwise Wilcoxon test and Benjamini–Hochberg method for $P$-value correction); ns, not significant. Underlying data for this figure are provided in https://doi.org/10.5281/zenodo.6537728. HEX, hexanol; IAA, isoamyl alcohol; sIAA, saturating IAA.

phenotype is that ODR-3 may be required for IAA-mediated saturation in AWC, leading to similar hexanol-evoked responses in *odr-3* mutants in both unsaturated and saturated conditions. Consistent with previous work showing that AWC retains partial ability to respond to IAA in *odr-3* mutants [41,65,67], IAA decreased intracellular calcium levels in AWC in *odr-3* animals, although both the response amplitude as well as the number of responding neurons were reduced as compared to responses in wild-type animals (S5B Fig). We conclude that hexanol attraction does not require ODR-3 but that ODR-3 is essential for hexanol aversion in sIAA. However, we are unable to exclude the possibility that IAA does not fully and/or bilaterally saturate AWC in *odr-3* mutants leading to the observed defect in hexanol-evoked activation of this neuron type.

## Hexanol-mediated inhibition but not activation of AWC requires the ODR-1 receptor guanylyl cyclase

We next asked whether hexanol acts via distinct downstream effector pathways in AWC in the absence or presence of sIAA to decrease or increase intracellular calcium levels, respectively. Both activation and inhibition by hexanol were abolished in animals mutant for the *tax-4* cyclic nucleotide-gated channel, indicating that both responses require cGMP signaling (Fig 4A). However, while *odr-1* mutants also failed to be attracted to hexanol and were instead weakly repelled (Fig 4B), these animals robustly avoided hexanol in sIAA (Fig 4B), indicating that ODR-1 is dispensable for hexanol avoidance, but is necessary for attraction.

Although ASH does not express *odr-1*, cGMP signaling in AWC has previously been shown to modulate ASH responses to a subset of nociceptive chemicals via a gap junction network [68]. In one model, hexanol responses could be lost in AWC in *odr-1* mutants regardless of odorant conditions, and hexanol avoidance could be driven by ASH alone. Alternatively, AWC may retain the ability to be activated but not inhibited by hexanol in *odr-1* mutants. To distinguish between these possibilities, we examined hexanol-evoked changes in AWC calcium dynamics in *odr-1* mutants. While IAA elicited responses of significantly reduced but variable amplitude, hexanol-evoked responses were largely abolished in AWC in *odr-1* mutants (Figs 4C and S4B and S5C), consistent with the inability of these animals to be attracted to either chemical [25,66,69]. However, in sIAA, hexanol again robustly increased intracellular calcium levels in AWC in *odr-1* mutants (Figs 4C and S5C), correlated with these mutants retaining the ability to avoid hexanol. These results indicate that while hexanol acts via ODR-1 to inhibit AWC and drive attraction in control conditions, in sIAA, hexanol acts via an ODR-1-independent pathway to activate these neurons, and promote aversion.

The DAF-11 receptor guanylyl cyclase is also necessary for sensory transduction in AWC and has been proposed to heterodimerize with ODR-1 [69–72]. We tested whether DAF-11 is required for hexanol-mediated inhibition or activation of AWC in control or sIAA conditions, respectively. Responses to hexanol were abolished in *daf-11(m47)* mutants (S5D Fig), similar to our observations in *odr-1* mutants (Fig 4C), indicating that both ODR-1 and DAF-11 are required for hexanol-evoked inhibition of AWC. In sIAA conditions, however, intracellular

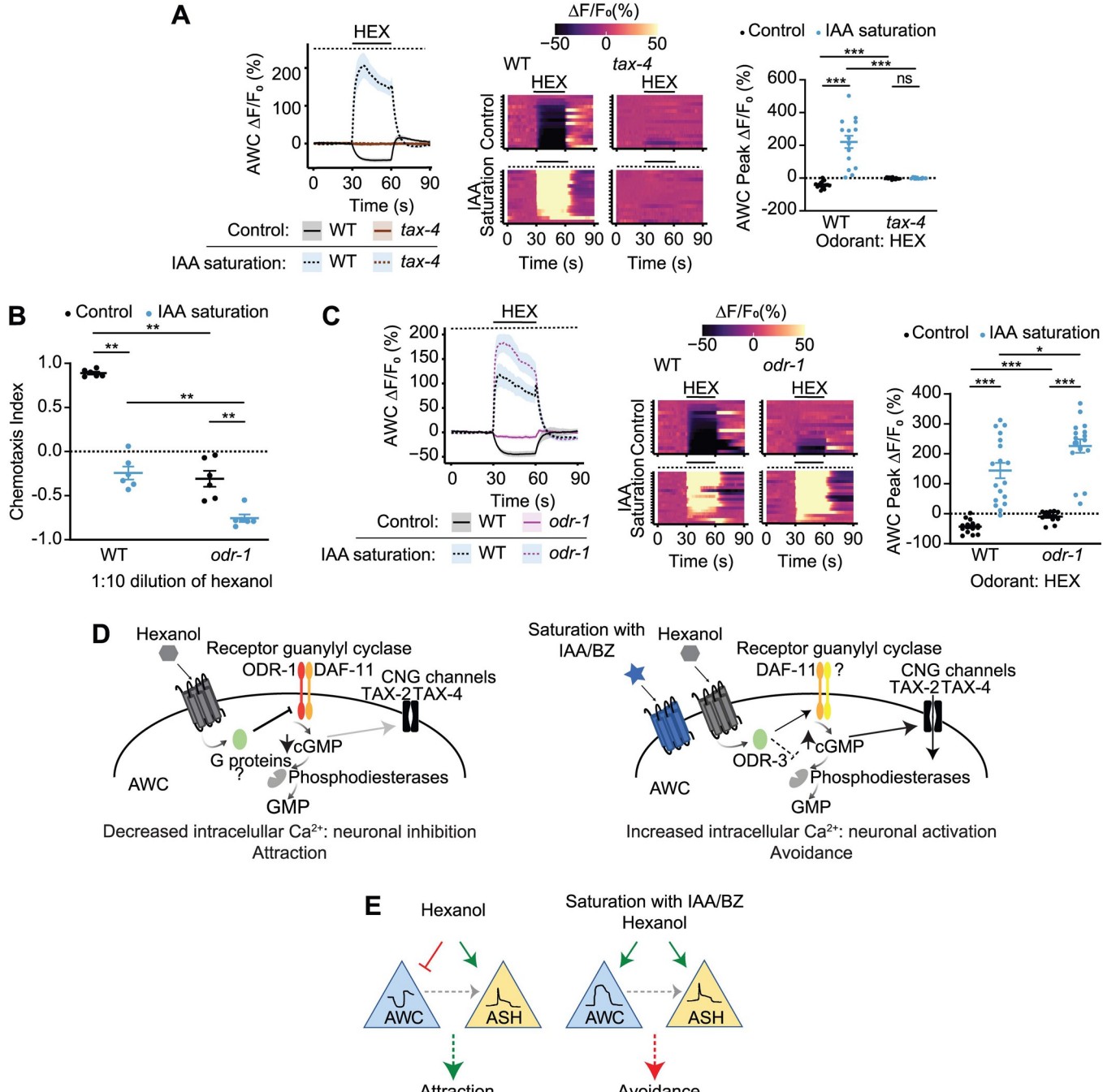

**Fig 4. The ODR-1 receptor guanylyl cyclase is required for hexanol-evoked inhibition but not activation of AWC. (A, C)** (Left) Average changes in GCaMP3 fluorescence in AWC in response to a pulse of $10^{-4}$ hexanol in wild-type, *tax-4(p678)*, and *odr-1(n1936)* animals. The presence of saturating chemicals in the imaging chip at $10^{-4}$ dilution is indicated by a dashed line. Shaded regions are SEM. (Center) Corresponding heatmaps of changes in fluorescence intensity are shown at right. Each row in the heatmaps shows responses from a single AWC neuron from different animals; $n \geq 15$ each. Control wild-type data were interleaved with experimental data in A and C; a subset of wild-type data in control conditions is repeated in these panels. A subset of wild-type data in control and sIAA conditions were also interleaved with experimental data in Fig 3D and are repeated. (Right) Quantification of fluorescence intensity changes upon hexanol onset under nonsaturated or saturated conditions. Each dot is the response from a single neuron. Raw traces of hexanol-evoked responses in AWC in control and sIAA conditions are shown in S5C Fig. **(B)** Behavioral responses of animals of the indicated genotypes to a point source of 1:10 dilution of hexanol on plates with or without sIAA at $10^{-4}$ dilution. Each dot is the chemotaxis index of a single assay plate containing approximately 100–200 adult hermaphrodites. Assays were performed in duplicate over at least 3 days. Long horizontal bars indicate the mean; errors are SEM. *, **, ***: $P < 0.05$, 0.01, and 0.001, respectively (Kruskal–Wallis with post hoc pairwise Wilcoxon test and Benjamini–Hochberg method for *P*-value correction). **(D)** Proposed model for hexanol-mediated inhibition and activation of AWC in an odorant context-dependent manner. See text for details. **(E)** In control conditions,

hexanol inhibits AWC, and AWC-driven attraction predominates over ASH-driven avoidance. In saturation conditions, hexanol activates both AWC and ASH; either neuron can drive avoidance. AWC may modulate ASH hexanol responses via peptidergic signaling (gray dashed arrow). Underlying data for this figure are provided in https://doi.org/10.5281/zenodo.6537728. BZ, benzaldehyde; HEX, hexanol; IAA, isoamyl alcohol; sIAA, saturating IAA.

calcium levels were again increased in a subset of *daf-11* mutants in response to hexanol, albeit at a lower frequency, than in wild-type animals (S5D Fig). Unlike *odr-1* mutants that retained partial responses to IAA (S5B Fig), IAA responses were largely abolished in *daf-11* animals (S5E Fig). We infer that while DAF-11 may be partly necessary for driving hexanol-evoked activation of AWC in sIAA, additional DAF-11- and ODR-1-independent pathways likely contribute to hexanol-evoked activation of AWC under these conditions (see Discussion).

## Discussion

Here, we show that the behavioral response of *C. elegans* to a food-related odor is inverted from attraction to avoidance in the continuous presence of a second attractive chemical. We find that this behavioral inversion is correlated with an inversion in the odorant response in a single olfactory neuron type via engagement of distinct intracellular signal transduction pathways in different chemical environments. Bidirectional responses of neurons such as parietal eye photoreceptors in lower vertebrates to blue or green light, olfactory neurons of *Drosophila* in response to different odors, and *C. elegans* sensory neurons to different concentrations of an odorant have been reported [19,67,73–76]. In this work, we describe a mechanism by which the same concentration of a chemical evokes bidirectional sensory responses in a context-dependent manner in a single chemosensory neuron type, and suggest that related principles may underlie aspects of stimulus encoding and stimulus discrimination across sensory modalities.

In control conditions, we propose that hexanol (and likely heptanol) acts via its cognate receptor(s) in AWC and Gα proteins other than or in addition to ODR-3 to inhibit the ODR-1 and DAF-11 receptor guanylyl cyclases to close the TAX-2/TAX-4 channels and decrease intracellular calcium concentrations (Fig 4D) [43]. However, in the presence of a subset of saturating AWC-sensed chemicals, the hexanol receptor instead acts via the ODR-3 Gα protein to increase cGMP levels and open the TAX-2/TAX-4 channels thereby increasing intracellular calcium to activate AWC (Fig 4D and 4E). AWC is predicted to express at least 6 receptor guanylyl cyclases including ODR-1 and DAF-11, as well as 5 phosphodiesterases [77]. ODR-3 may activate DAF-11 and additional guanylyl cyclases other than ODR-1, and/or inhibit one or more phosphodiesterases to increase intracellular cGMP levels in sIAA (Fig 4D). Our results suggest that odorant-mediated engagement of possibly different receptors and Gα proteins directs regulation of distinct downstream signaling pathways in AWC to drive context-dependent plasticity in neuronal and behavioral responses. This model also accounts for the ability of IAA to saturate AWC and switch the sign of the hexanol-evoked responses in AWC in *daf-11* mutants despite the failure of these neurons to respond to this odorant under control conditions. The engagement of distinct signaling pathways in different odorant contexts suggests that the observed increase in hexanol/heptanol-evoked intracellular calcium levels is unlikely to simply be due to disinhibition, but instead represents a stimulus-driven neuronal response. Hexanol-evoked inhibition of AWC likely overrides the response in ASH to drive robust attraction to low hexanol concentrations in control conditions, but in a subset of saturating odors, activation of either AWC or ASH is sufficient to drive avoidance of hexanol at these

concentrations (Fig 4E). Additional mechanisms may operate to drive avoidance of high concentrations of hexanol in control or sIAA conditions.

How might hexanol engage different downstream effector pathways under different odorant saturation conditions? Occupancy of a shared receptor by IAA may antagonize hexanol binding, and drive hexanol-mediated activation of a different signaling pathway via alternate AWC-expressed hexanol receptor(s). Antagonism of olfactory receptors by odorants in mixtures has now been extensively described and shown to play a role in stimulus encoding and odorant discrimination [15,17,78–81]. However, a mixture of IAA and hexanol does not activate AWC, and saturation with the structurally distinct chemical benzaldehyde is also sufficient to activate this neuron type. Although we are unable to exclude the possibility that a subset of AWC-sensed chemicals shares a broadly tuned receptor that alters neuronal responses based on odorant context [18,82], we favor the notion that saturation with IAA or benzaldehyde, but not diacetyl or pyrazine, alters neuronal state and signaling in a manner that then dictates the differential usage of intracellular signaling pathways by medium-chain alcohol receptor(s) to elicit distinct sensory responses. The as yet unidentified hexanol (and heptanol) receptor(s) may be modified under IAA or benzaldehyde saturation conditions to promote coupling to distinct effector pathways upon ligand binding [83–85]. Alternatively, differential compartmentalization of signaling complexes within the AWC sensory cilia membrane may promote the usage of distinct signal transduction machinery in different neuronal signaling conditions [66,86–89].

Odorants such as IAA and benzaldehyde are derived from amino acid degradation pathways in bacterial species that are preferred attractive food sources for *C. elegans* [90–93]. In contrast, hexanol is produced via chain elongation in bacteria that are likely to be poor food sources and/or pathogenic for *C. elegans* [94]. We speculate that *C. elegans* may be attracted to hexanol-producing bacteria under conditions where these bacteria are the only available food source. However, in the presence of odors indicating the presence of more nutritious bacteria, it may be advantageous for worms to instead avoid hexanol. The hexanol-evoked response inversion in AWC may be particularly critical to drive adaptive food choice behavior in the presence of other attractive odorants that are primarily sensed by AWC as compared to chemicals such as diacetyl and pyrazine, attraction to which is largely driven by other olfactory neuron types [25].

Behavioral plasticity driven by AWC and other sensory neurons has previously been shown to be largely regulated via modulation of synaptic transmission in the circuit, with little to no change in the primary sensory response [29,36,95–99]. A potential advantage of differential usage of intracellular signaling pathways over modulation of sensory neuron synaptic output is the ability to discriminate between, and differentially respond to, each stimulus sensed by that neuron in response to acute changes in odorant context. This mechanism is particularly advantageous for polymodal sensory neurons such as those in *C. elegans* [26] in which state-dependent engagement of different signaling pathways within a single sensory neuron type may allow animals to more effectively assess the salience of individual olfactory cues. Complex chemical response strategies have also been described in *Drosophila* gustatory neurons that express multiple ligand-gated ion channel receptors for different chemicals, and may represent a general mechanism by which organisms efficiently encode stimulus properties [100,101]. As the signaling content of sensory cells across different organisms is described more fully [77,102–105], a challenging next step will be to assess how different intra- and intercellular pathways are used under different conditions that animals encounter in the wild, and how this response flexibility is translated through the circuit to drive adaptive behavioral responses.

## Materials and methods

### Strains and growth conditions

All *C. elegans* strains were maintained on nematode growth medium (NGM) at 20˚C unless noted otherwise. Five days prior to behavioral assays, 10 L4 larvae per genotype were picked to 10 cm assay growth plates (day 1), and young adults were tested in behavioral and calcium imaging assays 4 days later (day 5). Animals were maintained under well-fed conditions at all times. To standardize growth conditions, NGM plates were seeded with bacteria as follows: concentrated *Escherichia coli* OP50 was cultured by inoculating 10 µl of a starter OP50 culture (grown in LB for approximately 2 hours from a single colony) per 1 L of SuperBroth media (3.2% w/v tryptone, 2.0% yeast extract, 0.5% NaCl). SuperBroth cultures grown overnight were treated with a low concentration of the antibiotic gentamicin (300 ng/ml; Sigma G1397) for approximately 4 hours, centrifuged for 20 minutes at 4˚C, and the resulting pellets resuspended in 75 ml of S-Basal buffer. The concentrated bacterial food was stored at −80˚C and thawed as needed to seed plates (1 ml/10 cm plate).

All strains were constructed using standard genetic procedures. The presence of mutations was confirmed by PCR-based amplification and/or sequencing. The *odr-1*p::*odr-3*::*SL2*::*mCherry* (PSAB1269) plasmid was injected at 10 ng/µl together with the *unc-122*p::*gfp* co-injection marker at 50 ng/µl to generate transgenic rescue strains. Expression patterns and phenotypes were confirmed in initial experiments using multiple independent transgenic lines, and a single line was selected for additional analysis. All calcium imaging experiments with the exception of those in *daf-11(m47)* were performed using a strain expressing GCaMP3 specifically in AWC from a stably integrated transgene (*oyIs91*; S1 Table). We were unable to generate a *daf-11(m47)* strain carrying *oyIs91* possibly due to genetic linkage of these alleles. Thus, calcium imaging in *daf-11* was performed using *odr-1*p::GCaMP3 expressed from extrachromosomal arrays. Transgenic animals were generated with the *odr-1*p:GCaMP3 (PSAB1201) plasmid injected at 20 ng/µl and the *unc-122*p:*mCherry* co-injection marker injected at 30 ng/µl. Animals from 2 independently obtained lines were examined.

A complete list of strains used in this work is provided in S1 Table.

### Molecular biology

An *odr-3* cDNA [62] was cloned into a worm expression plasmid containing approximately 1.0 kb upstream *odr-1* regulatory sequences using standard restriction enzyme cloning (PSAB1269).

### Plate chemotaxis assays

Chemotaxis assays were performed according to previously published protocols [25,106]. Assays were performed on 10 cm square or round plates with one or two 1 µl spots of odorant and the diluent ethanol at either end, together with 1 µl of 1 M sodium azide at each spot to immobilize worms. Odorants were diluted freshly in ethanol as needed. Saturation assays were performed using the same protocol, except that the relevant odorant was added to the assay agar before pouring plates (1 µl odorant at $10^{-4}$ dilution/10 ml agar). Animals were washed off growth plates with S-Basal and washed twice subsequently with S-Basal and once with Milli-Q water. Washed animals were placed at the center of the assay plate and allowed to move for an hour. The number of worms in 2 horizontal rows adjacent to the odor and ethanol spots was quantified at the end of the assay. Each assay was performed at

least in duplicate each day; data are reported from biologically independent assays performed on at least 3 days.

## Osmotic avoidance assay

Osmotic avoidance behavior assays were performed essentially as previously described [107,108]. Ten young adult worms were transferred without food to an agar plate and allowed to recover for at least 2 minutes. They were then placed in the center of an NGM plate with a ring of 8 M glycerol containing bromophenol blue (Sigma B0126). The number of worms inside and outside of the ring was counted after 10 minutes.

## Microfluidics behavioral assays

Microfluidics assays were performed following published protocols, using custom designed microfluidic devices (https://doi.org/10.5281/zenodo.6537728) [42]. The assembled microfluidic device was degassed in a vacuum desiccator for approximately 30 minutes prior to loading a 5% v/v poloxamer surfactant (Sigma P5556) with 2% xylene cyanol (2 mg/ml) solution through the outlet port. These steps ensured that the arena was bubble-free prior to loading the worms and stimulus reservoirs. Buffer and stimulus flowed by gravity from elevated reservoirs and were controlled with manual Luer valves. 20 to 30 young adult animals were transferred to unseeded plates and flooded with S-Basal buffer to remove any residual bacteria. The worms were then transferred into a tube and gently loaded into the buffer-filled arena via syringe. After allowing the worms to disperse throughout the arena (approximately 5 minutes), the flow of the odorant stimulus was started. Three parallel stripes flowed through the stimulus; 2 outer stripes consisted of buffer and the central stripe contained the odorant. A total of 2% xylene cyanol (2 mg/ml) was added to the odorant to allow visualization and tracking. For IAA saturation assays, in addition to the stimulus odorant in the middle stripe, $10^{-4}$ IAA was included in all buffer and stimulus reservoirs. Movies were recorded at 2 Hz on a PixelLink camera while worms were exposed to 20 minutes of constant odor. Following each experiment, the devices were flushed with water and soaked in ethanol overnight to remove any residual odorant. Prior to using the devices for additional assays, the chip was rinsed in water and baked at 50˚C for a minimum of 4 hours to evaporate any residual ethanol odor and liquid. The cleaning procedure was validated by buffer-buffer control assays, in which worms showed no spatial preference.

All movie acquisition, processing, and subsequent behavioral analysis was performed via modified custom MATLAB software [42] (https://doi.org/10.5281/zenodo.6537728). Data visualization and figures were generated using RStudio (version 1.3.959). A minimum of 3 assays per condition were performed on multiple days, and mean relative residency and chemotaxis index in respect to spatial stimuli was calculated. Briefly, the $y$-position data were binned into 50 bins for each assay. Relative residence was calculated by counting the # of tracks in each of the 50 $y$-position bins, and dividing each of these counts by the average # of counts across all 50 bins. Average residency histograms show the average of these residency values for each $y$-position bin over 3 assays/condition. Chemotaxis index was calculated as (normalized # of tracks within the odorant–normalized # of tracks in buffer)/total # of normalized tracks. Track numbers were normalized by calculating the # tracks X (total length of arena/length of respective buffer or odorant region). Stripe boundaries containing the odorant were determined using luminance data using xylene cyanol dye. To account for variable luminance across the device, luminance values were normalized using a linear regression fit. Boundaries were identified as the first and second sign switch of luminance values using the normalized luminance data.

## Calcium imaging

Calcium imaging was performed as previously described, using modified custom microfluidic devices [53,109]. Imaging was conducted on an Olympus BX52WI microscope with a 40× oil objective and Hamamatsu Orca CCD camera. Recordings were performed at 4 Hz. All odorants were diluted in S-Basal buffer and 1 µl of 20 µM fluorescein was added to one of the channels to confirm correct fluid flow. IAA saturation assays included IAA (1:10,000) in all channels, including the worm loading buffer. A total of 1 mM (−)-tetramisole hydrochloride (Sigma L9756) was added to the S-Basal buffer to paralyze body wall muscles and keep animals stationary. To prevent the chip from clogging, poloxamer surfactant (Sigma P5556) was also added to S-Basal while loading the worms. Odor-evoked calcium transients in AWC and ASH sensory neurons were similar in the presence or absence of these chemicals. AWC and ASH neurons were imaged for 1 cycle of 30-second buffer/30-second odor/30-second buffer stimulus. For preexposure experiments, animals were transferred to either control or IAA-saturated chemotaxis plates. After 1 hour on the chemotaxis plates, animals were loaded in the calcium imaging chip. Each animal was imaged within approximately 5 minutes after removal from the plate. Imaging was also performed in the presence of buffer only to ensure that observed neuronal responses were to the odor stimulus and not artifactual.

Recorded image stacks were aligned with Fiji using the Template Matching plugin, and cropped to a region containing the cell body. The region of interest (ROI) was defined by outlining the desired cell body; background subtracted fluorescence intensity of the ROI was used for subsequent analysis. To correct for photobleaching, an exponential decay was fit to fluorescence intensity values for the first 30 seconds and the last 20 seconds of imaging (prior and post stimulus). The resulting curve was subtracted from original intensity values. Amplitude was calculated as maximum change in fluorescence (F-$F_0$) in the 10 seconds following odor addition; $F_0$ was set to the average $\Delta F/F_0$ value for 5 seconds before odor onset. Data visualization and figures were generated using RStudio (version 1.3.959). Photomask designs for customized microfluidic olfactory chips adapted from are available [53] (https://doi.org/10.5281/zenodo.6537728). Reported data were collected from biologically independent experiments over at least 2 days.

## Statistical analyses

Excel (Microsoft) and GraphPad Prism version 9.0.2 (www.graphpadpad.com) were used to generate all chemotaxis plate assay data. Plate chemotaxis index and peak $\Delta F/F_0$ amplitude data comparing 2 groups were analyzed using the Mann–Whitney–Wilcoxon test; data comparing more than 2 groups were analyzed using the Kruskal–Wallis test followed by the post hoc pairwise Wilcoxon test and Benjamini–Hochberg method for $P$-value correction. Since the chemotaxis index from microfluidics behavioral assays is derived from continuous distributional data of animal position over the duration of the assay, these data were analyzed using 2-way ANOVA followed by Bonferroni's multiple comparison test. All statistical analyses were performed in R (https://www.R-project.org/) and RStudio (http://www.rstudio.com)' and GraphPad Prism version 9.0.2 (www.graphpadpad.com). The tests used are indicated in the corresponding figure legends.

## Supporting information

**S1 Fig. Attraction to hexanol is switched to aversion in sIAA.** (A) Behaviors of wild-type animals on control plates or plates saturated with either hexanol or 1-heptanol (at $10^{-4}$ dilution). Test odorant: 1:1,000 dilution of IAA. (B) Behaviors of wild-type animals on control or sIAA plates to the indicated concentrations of hexanol. (C) Average residence histograms and

heatmaps as described for Fig 1D. $n$ = 20–30 animals per assay; 3 biological replicates. Test odorant: 1:9,000 dilution of IAA. **(D)** Chemotaxis indices calculated from behavioral assays shown in **C**. **(E)** Behaviors of animals of the indicated genotypes on control or sIAA plates to undiluted hexanol. The *odr-1* promoter was used to drive *unc-103(gf)* in AWC [110]. **(F)** Behaviors of animals of the indicated genotypes to a 1:200 dilution of benzaldehyde. **(G)** Shown is the percentage of animals of the indicated genotypes that escaped a ring of 8 M glycerol. In A, E, F, each dot is the chemotaxis index of a single assay plate containing approximately 100–200 adult hermaphrodites. Assays were performed in duplicate on at least 3 days. In B, each dot is the average chemotaxis index of 3–4 independent assays of approximately 100–200 animals each. In D, each dot is the chemotaxis index from a single assay in microfluidics chips. In G, each dot represents a single osmotic avoidance assay of 10 animals. Long horizontal bars indicate the mean; errors are SEM. *, **, ***: $P < 0.05$, 0.01, and $P < 0.001$, respectively (A, B, E, Kruskal–Wallis with post hoc pairwise Wilcoxon test and Benjamini–Hochberg method for $P$-value correction, F, G, Mann–Whitney–Wilcoxon test, D, 2-way ANOVA with Bonferroni's correction). Underlying data are provided in https://doi.org/10.5281/zenodo.6537728. IAA, isoamyl alcohol; sIAA, saturating IAA.
(EPS)

**S2 Fig. Hexanol-mediated inhibition and activation of AWC may be largely cell-autonomous. (A)** Behaviors of wild-type animals or animals expressing GCaMP3 under the *odr-1* promoter in AWC to a point source of 1:10 dilution of hexanol on plates with or without sIAA at $10^{-4}$ dilution. Each dot is the chemotaxis index of a single assay plate containing approximately 100–200 adult hermaphrodites. Assays were performed in duplicate over at least 3 days. Long horizontal bars indicate the mean; errors are SEM. **: $P < 0.01$ (Kruskal–Wallis with post hoc pairwise Wilcoxon test and Benjamini–Hochberg method for $P$-value correction); ns, not significant. **(B, D–H)** (Left) Average changes in GCaMP3 fluorescence in AWC in response to a pulse of the indicated odorants (solid line). The presence of saturating chemicals in the imaging chip at $10^{-4}$ dilution is indicated by a dashed line. Concentrations of test odorants used were $10^{-4}$ unless indicated otherwise. In F, animals were preexposed to a $10^{-4}$ dilution of IAA (see Methods). Shaded regions are SEM. (Center in B,D,G, Right in E,F,H) Corresponding heatmaps of changes in fluorescence intensity. Each row in the heatmaps shows responses from a single AWC neuron from different animals; $n \geq 14$ each. (Right in B, D,G) Quantification of peak fluorescence intensity changes upon odorant onset under nonsaturated or saturated conditions. Each dot represents the response from a single neuron. Control wild-type IAA response data were interleaved with experimental data in B and G and are repeated. Wild-type control responses to hexanol in S2E Fig are interleaved with experimental data in Fig 2A and 2E, and a subset of data in Fig 3D and are repeated. Long horizontal bars indicate the mean; errors are SEM. ***: $P < 0.001$ (B,G, Mann–Whitney–Wilcoxon test, D, Kruskal–Wallis with post hoc pairwise Wilcoxon test and Benjamini–Hochberg method for $P$-value correction). Quantification of responses in H are shown in Fig 2B. Alleles in H were *unc-13(e51)* and *unc-31(e928)*. **(C)** Raw traces of fluorescence changes in AWC in response to $10^{-4}$ hexanol in control and sIAA conditions corresponding to data shown in Fig 2A. Underlying data are provided in https://doi.org/10.5281/zenodo.6537728. HEX, hexanol; IAA, isoamyl alcohol; sIAA, saturating IAA.
(EPS)

**S3 Fig. Context-dependent response inversion in AWC is specific to particular odorant combinations. (A, B, D, E)** (Left) Average changes in GCaMP3 fluorescence in AWC in response to a pulse of the indicated odorants (solid line). The presence of saturating chemicals in the imaging chip is indicated by a dashed line. Shaded regions are SEM. Concentrations of

saturating and test chemicals: IAA at $10^{-4}$, diacetyl (DIA) at $10^{-3}$, pyrazine (PYR) at 10 mM. (Center) Corresponding heatmaps of changes in fluorescence intensity. Each row in the heatmaps shows responses from a single AWC neuron from different animals; $n \geq 10$ each. (Right) Quantification of peak fluorescence intensity changes upon odorant onset under non-saturated or saturated conditions. Each dot represents the response from a single neuron. Long horizontal bars indicate the mean; errors are SEM. *, ***: $P < 0.05$ and $<0.001$, (Mann–Whitney–Wilcoxon test); ns, not significant. **(C)** Behaviors of wild-type animals to a point source of 10 mg/ml of pyrazine on plates with or without sIAA at $10^{-4}$ dilution. Each dot is the chemotaxis index of a single assay plate containing approximately 100–200 adult hermaphrodites. Assays were performed in duplicate over at least 3 days. Long horizontal bars indicate the mean; errors are SEM. ns, not significant (Mann–Whitney–Wilcoxon test). **(F)** Summary table of test odorants and saturating odorant combinations and corresponding responses in AWC. Underlying data are provided in https://doi.org/10.5281/zenodo.6537728. Act; activation; IAA, isoamyl alcohol; Inh, inhibition; sIAA, saturating IAA; –, no response. (EPS)

**S4 Fig. ASH responds partly cell-autonomously to hexanol. (A)** Behaviors of wild-type or a strain expressing GCaMP3 in ASH under the *sra-6* promoter to a point source of undiluted hexanol on plates without or with sIAA at $10^{-4}$ dilution. Each dot is the chemotaxis index of a single assay plate containing approximately 100–200 adult hermaphrodites. Assays were performed in duplicate over at least 3 days. Long horizontal bars indicate the mean; errors are SEM. **: $P < 0.01$ (Kruskal–Wallis with post hoc pairwise Wilcoxon test and Benjamini–Hochberg method for $P$-value correction); ns, not significant. **(B–E)** (Top) Average changes in GCaMP3 fluorescence in ASH in response to a pulse of $10^{-4}$ dilution of hexanol (solid line) in animals of the indicated genetic backgrounds. The presence of saturating chemicals in the imaging chip at $10^{-4}$ dilution is indicated by a dashed line. Shaded regions are SEM. (Bottom) Corresponding heatmaps of changes in fluorescence intensity. Each row in the heatmaps shows responses from a single ASH neuron from different animals. $n = 10$ (*tax-4*), 15 (all other genotypes). Alleles used were *unc-13(e51)*, *unc-31(e928)*, and *tax-4(p678)*. Quantification of peak response amplitudes is shown in Fig 2H. Underlying data are provided in https://doi.org/10.5281/zenodo.6537728. sIAA, saturating IAA. (EPS)

**S5 Fig. Hexanol acts via distinct signaling pathways to inhibit or activate AWC in a context-dependent manner. (A)** Behaviors of animals of the indicated genotypes (see S1 Table) to a point source of 1:10 dilution of hexanol on plates with or without sIAA at $10^{-4}$ dilution. Each dot is the chemotaxis index of a single assay plate containing approximately 100–200 adult hermaphrodites. Assays were performed in duplicate over at least 3 days. Long horizontal bars indicate the mean; errors are SEM. ***: $P < 0.001$ (Kruskal–Wallis with post hoc pairwise Wilcoxon test and Benjamini–Hochberg method for $P$-value correction); ns, not significant. **(B, D, E)** (Left) Average changes in GCaMP3 fluorescence in AWC in response to a 30-second pulse of $10^{-4}$ dilution of IAA or hexanol (solid line) in animals of the indicated genetic backgrounds. The presence of saturating chemicals in the imaging chip at $10^{-4}$ dilution is indicated by a dashed line. Alleles used were *odr-1(n1936)*, *odr-3(n2150)*, and *daf-11(m47)*. Shaded regions are SEM. (Center) Corresponding heatmaps of changes in fluorescence intensity. Each row in the heatmaps shows responses from a single AWC neuron from different animals; $n \geq 15$ each. (Right) Quantification of peak response amplitudes. Each dot is the response from a single neuron. Long horizontal bars indicate the mean; errors are SEM. *, **, ***: $P < 0.05$, 0.01, and 0.001, respectively (B, D, Kruskal–Wallis with post hoc pairwise Wilcoxon test and Benjamini–Hochberg method for $P$-value correction; E, Mann–Whitney–Wilcoxon

test); ns, not significant. **(C)** Raw traces of fluorescence changes in AWC in animals of the indicated genotypes in response to $10^{-4}$ hexanol in control and sIAA conditions corresponding to data shown in Figs 3D and 4C. Underlying data are provided in https://doi.org/10.5281/zenodo.6537728. sIAA, saturating IAA.

(EPS)

**S1 Video. Wild-type animals are attracted to, or avoid, hexanol in the absence or presence of sIAA in microfluidics behavioral arenas.** Video shows the locomotor responses of wild-type worms to odor stripes of $10^{-4}$ hexanol (HEX) in control (left) and sIAA (HEX/sIAA, right) conditions in a microfluidics behavioral chip. Odor stripes contain dye for visualization. Video is accelerated 40×.

(MP4)

**S2 Video. Hexanol decreases or increases intracellular calcium in AWC in the absence or presence of sIAA, respectively.** A wild-type animal expressing the *odr-1*p::*GCaMP3* transgene was subjected to a 30-second pulse of $10^{-4}$ hexanol (HEX onset) in control (HEX, left) and sIAA (HEX/sIAA, right) conditions. Video is accelerated 6×.

(MOV)

**S1 Table. Strains used in this work.**
(DOCX)

## Acknowledgments

We are grateful to the *Caenorhabditis* Genetics Center and Steven Flavell, Yun Zhang, and Gert Jansen for providing strains, the Brandeis Materials Research Science and Engineering Center (MRSEC) for access to the microfabrication facility, Dirk Albrecht and Till Hartmann for advice with analyses of microfluidics behavioral assays, and Ashish Maurya for experimental assistance. We thank members of the Sengupta lab and Oliver Hobert for critical comments on the manuscript.

## Author Contributions

**Conceptualization:** Anna H. Hartmann, Noelle D. Dwyer, Cornelia I. Bargmann, Piali Sengupta.

**Data curation:** Munzareen Khan, Michael P. O'Donnell.

**Formal analysis:** Munzareen Khan, Michael P. O'Donnell.

**Funding acquisition:** Piali Sengupta.

**Investigation:** Munzareen Khan, Michael P. O'Donnell, Madeline Piccione, Anjali Pandey, Pin-Hao Chao.

**Methodology:** Munzareen Khan, Anna H. Hartmann.

**Project administration:** Piali Sengupta.

**Software:** Anna H. Hartmann, Michael P. O'Donnell.

**Supervision:** Piali Sengupta.

**Validation:** Munzareen Khan.

**Visualization:** Munzareen Khan, Anna H. Hartmann, Michael P. O'Donnell.

**Writing – original draft:** Munzareen Khan, Piali Sengupta.

**Writing – review & editing:** Munzareen Khan, Anna H. Hartmann, Michael P. O'Donnell, Noelle D. Dwyer, Cornelia I. Bargmann, Piali Sengupta.

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
