## [Editor Report · Decision Letter 0]

16 Nov 2021

Dear Piali, 

Thank you for submitting your manuscript entitled "Context-dependent inversion of the response in a single sensory neuron type reverses olfactory preference behavior" for consideration as a Short Reports by PLOS Biology. Apologies for our delay in sending you an initial decision - we wished to discuss your manuscript with an Academic Editor, and had a bit of trouble finding someone who was available to provide advice last week.

We have now had a chance to discuss your submission with an academic editor with relevant expertise, and I am writing to let you know that we would like to send your submission out for external peer review.

Once your full submission is complete, your paper will undergo a series of checks in preparation for peer review. Once your manuscript has passed the checks it will be sent out for review. 

As a note, if your manuscript has been previously reviewed at another journal, PLOS Biology is willing to work with those reviews in order to avoid re-starting the process. Submission of the previous reviews is entirely optional and our ability to use them effectively will depend on the willingness of the previous journal to confirm the content of the reports and share the reviewer identities. Please note that we reserve the right to invite additional reviewers if we consider that additional/independent reviewers are needed, although we aim to avoid this as far as possible. In our experience, working with previous reviews does save time. 

If you would like to send your previous reviewer reports to us, please specify this in the cover letter, mentioning the name of the previous journal and the manuscript ID the study was given, and include a point-by-point response to reviewers that details how you have or plan to address the reviewers' concerns. Please contact me at the email that can be found below my signature if you have questions. 

Please re-submit your manuscript within two working days, i.e. by Nov 18 2021 11:59PM.

Kind regards,

Lucas

Lucas Smith

Associate Editor

PLOS Biology

lsmith@plos.org

---

## [Decision Letter · Decision Letter 1]

5 Jan 2022

Dear Piali,

Thank you for submitting your manuscript "Context-dependent inversion of the response in a single sensory neuron type reverses olfactory preference behavior" for consideration as a Short Report at PLOS Biology, and apologies again for our delay in sending you a decision. Your manuscript has been evaluated by the PLOS Biology editors, an Academic Editor with relevant expertise, and by several independent reviewers.

The reviews are appended below. As you will see, all three reviewers find the conclusions of your study interesting, however they have also raised a number of technical issues and suggest additional experiments and analysis which would be needed to strengthen the study. In particular, Reviewers 2 and 3 think that it would be important for you to provide additional mechanistic insights into how these effects are mediated. While we appreciate that the study was submitted as a Short Report which can be somewhat preliminary, given the reviewer comments and after discussion with the Academic Editor, we think that it would be important to experimentally address these concerns and provide further insights into which receptors are responsible for the switch reported here and examine the role of cGMP. We think that this may require expanding the study into our longer Research Article format.

As a note, while the reviewers have also raised the important point that the ecological significance of this phenomenon is yet unclear, we would not require additional experiments or data to elucidate this aspect. We think you could address this point by adding discussion on if/how similar phenomenon might be encountered in nature and of the broader implications of these findings.

In light of the reviews, we will not be able to accept the current version of the manuscript, but we would welcome re-submission of a much-revised version that takes into account the reviewers' comments. We cannot make any decision about publication until we have seen the revised manuscript and your response to the reviewers' comments. Your revised manuscript is also likely to be sent for further evaluation by the reviewers.

We expect to receive your revised manuscript within 3 months. 

**IMPORTANT - SUBMITTING YOUR REVISION**

*Re-submission Checklist*

*Published Peer Review*

*PLOS Data Policy*

*Blot and Gel Data Policy*

Sincerely,

Lucas Smith

Associate Editor

PLOS Biology

lsmith@plos.org

REVIEWS:

Reviewer #1: Khan et al. investigate how saturation with one odorant changes the perception of another odorant from attraction to repulsion. This change is shown to occur at the level of chemotaxis behavior and odorant encoding by calcium levels in a specific olfactory neuron. Specific genes acting within the olfactory neuron are required for these changes, providing key initial components of a mechanism operating within an olfactory neuron. Together, the correlated changes in behavior and olfactory neuron activity make a good case for the conclusions in the manuscript. At a broader level, this work provides additional insights into how complex olfactory environments are interpreted and encoded to generate navigational behavior.

Throughout the paper, the data were very clearly visualized at both coarse- and fine-grained levels, making the results easy to understand and convincing. Many of the chemotaxis assays were also rigorously repeated on plates and in microfluidic devices.

Major issues:

1. Calcium imaging is quantified by (F-F0) / F0. Formally, the reported changes could be due to F or F0. Normally it is assumed that F0 is constant, and changes are due to F. However, odorant saturation would be expected to change F0. This would affect comparisons between the presence and absence of saturation. 

I agree with reporting the results as (F-F0) / F0 because it is most relevant to show on the changes from the F0 baseline since it reveals whether the odorant is changing the activity of the neuron. 

Nonetheless, there might be conditions where a flip between increased and decreased (F-F0) / F0 are due to changes in F0 when during IAA saturation, rather than reflecting the total activity of the neuron. For example, if the drop in F0 caused by the presence of IAA dominates, then (F-F0) / F0 could flip from negative to positive.

This issue could be addressed by examining the F0 values with and without IAA saturation within existing data, which could help refine the interpretations of the calcium imaging without the need to add any additional experiments. Also, if there is a prior literature on whether the targets of AWC relevant to chemotaxis sense change or absolute levels, the information would also help resolve the issue.

2. It is counterintuitive that loss of odr-1 leads to increased calcium, since odr-1 is predicted to activate TAX-2/TAX-4. Is it because odr-1 mutants have a lower baseline calcium (related 1 above)? I suggest that the authors address this in the discussion.

Minor issues:

1. Page 6: "We infer that distinct underlying antagonistic neuronal pathways mediate attraction to, and avoidance of, hexanol at all concentrations. Attraction predominates at lower, and avoidance at higher, hexanol concentrations; sIAA inhibits the attraction but not the avoidance pathway."

The line of argument in this section requires some clarification. Is this setting up the hypothesis that the mechanism for switching to avoidance for high hexanol is the same as for switching to hexanol avoidance under sIAA? 

It also appears that the authors are suggesting that the avoidance mechanisms for octanol, high hexanol, and hexanol under sIAA are the same, and trying to provide a single unified mechanism for three subtly different behaviors. I suggest that the discussion is a better place for this, since it does not really set up the precise question that authors address in subsequent sections.

2. page 9: "...in unc-13 and unc-31 mutants that lack synaptic and peptidergic transmission, respectively (Sieburth et al., 2007; Speese et al., 2007)."

This statement is inaccurate and should be revised. Sieburth 2007 shows that unc-13 has a peptidergic secretion defect that is just as strong as unc-31. Sieburth 2007 states in the abstract: "Similar neuropeptide secretion defects were found in mutants lacking unc-31 (encoding the protein CAPS) or unc-13 (encoding Munc13)."

3. The use of unc-13 and unc-31 to test cell-autonomy are good experiments, but these mutations do not affect signalling via gap junctions. The potential for gap junctions to play a role should be stated as a caveat.

4. In the section on testing the role of ASH, it seems to be implied that tax-4 affects AWC but not ASH. This should be clarified by stating whether tax-4 is expressed in ASH.

5. The figure or figure legends for 1G should indicate the genotypes of the AWC- and ASH- animals. Presumably these are the caspase strains listed in Table S1? Or were these laser-ablated?

6. Different statistical tests and corrections for multiple hypotheses testing were used for similar data types. In the chemotaxis assays, Mann Whitney Wilcoxon or Kruskal-Wallis with posthoc pairwise Wilcox and Benjamin-Hochberg corrections were used for one set, and the two-way ANOVA with Bonferroni for another. The rationale for using different tests instead of 1 consistent test should be justified in the methods. If not, please use the same test for similar data types.

7. For ease of comparison among the many calcium imaging experiments in Figures 2 and 4 where the panels have similar sets of data in a similar order, I suggest:

(i) aligning them in portrait orientation; and

(ii) using consistent labelling by including the name of the neuron in all Y-axes labels and removing the odorant from the Y-axes since they are redundant with the stimulus label.

Reviewer #2: The manuscript reports an interesting phenomenon whereby the response to a few specific odorants switches from inhibition to activation in the presence of a constant source of another odorant. There is a concomitant change in the behavioral response, going from attraction to aversion in the presense of the background odorant. The reversal in Ca response in the AWC neuron as well as the behavioral response are gone in the odr-3 mutant. 

While there is much to appreciate in this study, the main concerns that emerge are listed below:

1. The GPCR odorant receptors that detect the odorants involved in this study are yet to be identified. In the absence of this information the authors are unabvle to test the mechanism of response reversal adequately. Could it be competitive or allosteric interactions at the same receptor? Or some form of integrative effect downstream of two different receptors for the 2 odorants? Is there a receptor specific adaptation or modification? Without the ability to do such experiments the analyses of the phenomenon using downstream factors seems a bit preliminary. 

2. The ecological role of such a phenomenon is unclear and not tested by the authors. What could be the benefit of switching from inhibition (attraction) to excitation (aversion) for the one odorant in the presence of IAA is unclear?

3. The authors report the Delta F/F values in the plots showing the activities of AWC, however we get little sense for whether the baseline levels of Ca in the AWC neurons are dramatically different or not. Also, do these neurons fire action potentials like the AWA and if so, it would be useful to also find out the changes in the baseline and stimulus dependent firing rate? 

4. Since this is a unusual phenomenon, understanding it in more detail, such as across multiple doses of the sIAA and hexanol, or across a larger panel of odorants, would have been useful. For example , based on the few chemical structures tested it would appear that the two classes of odorants use distinct binding sites or distinct reporters. Would that interpretation be supported from additional odorants tested? 

Reviewer #3: Since the early 1990s, studies of the neural, genetic, and behavioral mechanisms of C. elegans olfactory behavior have been instrumental in building the "modern synthesis" that guides our understanding of worm neurobiology. Here, Khan et al build on intriguing findings from early studies by Cori Bargmann showing that single pairs of worm olfactory neurons can mediate behavioral responses to multiple odorants, even when one of these odorants is present at saturating concentrations. Khan et al show, quite remarkably, that AWC-sensed medium-chain alcohols like hexane (HEX) and heptane (HEX), which typically elicit attraction, become repellants when animals encounter them in the presence of another odorant that saturates AWC responses, especially isoamyl alcohol (IAA). The authors compellingly show that these inverted behavioral responses are driven by the inversion of AWC's physiological response to HEX and HEPT, which now increase AWC Ca++ rather than decrease it. Moreover, they find that this inversion is an AWC-autonomous process: while the nociceptive neuron ASH also has a role in HEX and HEPT aversion, the change in behavior results primarily from a cell-autonomous change in AWC physiology. They go on to dissect some of the components required for this plasticity, showing that HEX and HEPT appear to engage different G-protein signaling pathways depending on whether saturating IAA is present.

The description of this novel form of plasticity is exciting: it reveals a new situation in which sensory context can dramatically change innate behavioral responses. Further, the mechanistic insight provided (AWC autonomy, inversion of physiological responses, different roles of G proteins) is an important advance. However, there are some issues that somewhat reduce my enthusiasm. Most important of these is that the paper concludes without identifying the mechanism by which sIAA alters AWC physiology to reconfigure HEX/HEPT reponses. As a Short Report, this might not be a concern, but I'm sure whether that the level of general interest here is high enough to compensate for that.

One possibility that seems worth exploring more deeply would be the role of cGMP. Under sIAA conditions, [cGMP] in AWC might be quite low - I wonder whether artificially lowering [cGMP] under control conditions might cause HEX/HEPT to become repellants. A related issue is about the role of the RGC odr-1 - it's fascinating that loss of odr-1 nearly recapitulates the effect of sIAA with respect to HEX behavior and pushes the Ca++ response in the same direction. Maybe the Ca++ response to HEX would invert (and generate stronger HEX repulsion) in daf-11 or daf-11; odr-1 animals? I wonder whether somehow increasing [cGMP] in AWC might suppress the odr-1 HEX-aversion phenotype (and/or the effect of sIAA). Maybe odr-1 mutants are defective for IAA attraction because the normal IAA signaling pathway is already being pushed towards "saturation"?

Minor issues

1. Is there any reason to believe that worms might encounter persistent, saturating concentrations of AWC-sensed odorants in the wild? The idea that this mechanism could help animals distinguish between food sources is interesting, but it also seems possible (even likely) that that the behavioral inversion seen in the lab is the result of pushing the system beyond its normal dynamic range. That doesn't make these findings uninteresting - they tell us something important about how AWC works - but they limit the potential that the phenomenon identified here has any adaptive value.

2. Fig 1C-E - please indicate that the odorant is HEX

3. Fig S2B clearly shows that AWC can still somehow see changes in IAA concentration in sIAA. What's the significance of this? How can it be reconciled with the lack of IAA attraction in sIAA?

4. The response of AWC to 2-methylpyrazine (Fig S2E) seems surprising given, as the authors note, that this compound is thought to be detected primarily by AWA. It would be useful for the authors to comment on the potential significance of this result.

5. Fig 3C: Please add an "odorant HEX" label

6. p. 13: "A trivial explanation for the observed phenotype is that ODR-3 may be required for IAA-mediated saturation in AWC, leading to similar hexanol-evoked responses in odr-3 mutants in both unsaturated and saturated conditions." This may be a straightforward possibility, but I don't think it's a trivial one. It would still provide important insight into the diversity of signaling mechanisms in AWC.

7. p. 13: Please add "...in odr-3 mutants" to "Consistent with previous work showing that AWC retains partial ability to respond to IAA"

---

## [Editor Report · Decision Letter 2]

27 Apr 2022

Dear Piali,

Thank you for submitting your revised Short Report entitled "Context-dependent inversion of the response in a single sensory neuron type reverses olfactory preference behavior" for publication in PLOS Biology. I have now obtained advice from the Academic Editor and I am pleased to say that we are satisfied by the changes made with the revision. However, before we can editorially accept the study, we ask that you please address the following two points in a revised manuscript:

1) DATA SHARING: Thank you for providing the data underlying your figures on github ( https://github.com/SenguptaLab/HEXplasticity.git). Unfortunately, this type of deposition is not fully compliant with our data sharing policy, as the GitHub deposition does not offer sufficient version control/provide a DOI. Therefore, we request that you please use Zenodo to archive your data repository posted on GitHub. For more information on how to do this, see: https://docs.github.com/en/repositories/archiving-a-github-repository/referencing-and-citing-content

Alternatively, you can supply the underlying data as supplementary files to be published with your manuscript.

For more information on PLOS’ data policy see http://journals.plos.org/plosbiology/s/data-availability

2) TITLE: After discussion with the team, we feel the title of your piece might be edited slightly to improve clarity. If you agree, we suggest changing it to something like: “Context-dependent reversal of odorant preference is driven by inversion of the response in a single sensory neuron type”

We expect to receive your revised manuscript within two weeks. 

*Published Peer Review History*

*Press*

Sincerely,

Lucas

Lucas Smith, Ph.D.,

Associate Editor,

lsmith@plos.org,

PLOS Biology

---

## [Editor Report · Decision Letter 3]

16 May 2022

Dear Piali,

On behalf of my colleagues and the Academic Editor, Claude Desplan, I am pleased to say that we can in principle accept your Short Report, "Context-dependent reversal of odorant preference is driven by inversion of the response in a single sensory neuron type" for publication in PLOS Biology, provided you address any remaining formatting and reporting issues. These will be detailed in an email that will follow this letter and that you will usually receive within 2-3 business days, during which time no action is required from you. Please note that we will not be able to formally accept your manuscript and schedule it for publication until you have completed any requested changes.

PRESS

Sincerely, 

Lucas Smith, Ph.D. 

Senior Editor 

PLOS Biology

lsmith@plos.org